# Genome-Based Analysis of *Verticillium* Polyketide Synthase Gene Clusters

**DOI:** 10.3390/biology11091252

**Published:** 2022-08-23

**Authors:** Mohammad Sayari, Aria Dolatabadian, Mohamed El-Shetehy, Pawanpuneet Kaur Rehal, Fouad Daayf

**Affiliations:** 1Department of Plant Science, Faculty of Agricultural and Food Sciences, University of Manitoba, 222 Agriculture Building, Winnipeg, MB R3T 2N2, Canada; 2Department of Botany, Faculty of Science, Tanta University, Tanta 31527, Egypt

**Keywords:** *Verticillium*, PKS, cluster, genomics, phylogeny

## Abstract

**Simple Summary:**

Fungi can produce many types of secondary metabolites, including mycotoxins. Poisonous mushrooms and mycotoxins that cause food spoilage have been known for a very long time. For example, *Aspergillus flavus*, which can grow on grains and nuts, produces highly toxic substances called Aflatoxins. Despite their menace to other living organisms, mycotoxins can be used for medicinal purposes, i.e., as antibiotics, growth-promoting compounds, and other kinds of drugs. These and other secondary metabolites produced by plant-pathogenic fungi may cause host plants to display disease symptoms and may play a substantial role in disease progression. Therefore, the identification and characterization of the genes involved in their biosynthesis are essential for understanding the molecular mechanism involved in their biosynthetic pathways and further promoting sustainable knowledge-based crop production.

**Abstract:**

Polyketides are structurally diverse and physiologically active secondary metabolites produced by many organisms, including fungi. The biosynthesis of polyketides from acyl-CoA thioesters is catalyzed by polyketide synthases, PKSs. Polyketides play roles including in cell protection against oxidative stress, non-constitutive (toxic) roles in cell membranes, and promoting the survival of the host organisms. The genus *Verticillium* comprises many species that affect a wide range of organisms including plants, insects, and other fungi. Many are known as causal agents of *Verticillium* wilt diseases in plants. In this study, a comparative genomics approach involving several *Verticillium* species led us to evaluate the potential of *Verticillium* species for producing polyketides and to identify putative polyketide biosynthesis gene clusters. The next step was to characterize them and predict the types of polyketide compounds they might produce. We used publicly available sequences from ten species of *Verticillium* including *V. dahliae*, *V. longisporum*, *V. nonalfalfae*, *V. alfalfae*, *V. nubilum*, *V. zaregamsianum*, *V. klebahnii*, *V. tricorpus*, *V. isaacii*, and *V. albo-atrum* to identify and characterize PKS gene clusters by utilizing a range of bioinformatic and phylogenetic approaches. We found 32 putative PKS genes and possible clusters in the genomes of *Verticillium* species. All the clusters appear to be complete and functional. In addition, at least five clusters including putative DHN-melanin-, cytochalasin-, fusarielien-, fujikurin-, and lijiquinone-like compounds may belong to the active PKS repertoire of *Verticillium*. These results will pave the way for further functional studies to understand the role of these clusters.

## 1. Introduction

*Verticillium* spp. are ascomycete fungi with an extensive host range of more than 350 plant species [1,2]. Out of ten main plant-associated species, six cause vascular wilts in crops [3], with *V. dahliae* and *V. albo-atrum* being two of the most destructive, responsible for billions of dollars in annual losses worldwide [1,4]. Among these, *V. dahliae* is the most ubiquitous species and the primary cause of *Verticillium* wilt, especially in temperate and subtropical climates [1]. 

Controlling pathogenic *Verticillium* species is difficult due to their melanized resting structures known as microsclerotia, which serve as the primary inoculum and survive in the soil for years [5]. In addition, *Verticillium* sp. populations are highly diverse, with variable and complex pathogenicity due to frequent chromosomal rearrangements and the involvement of many pathogenicity-related genes [6]. The latter have been studied in *V. dahliae* using genome, transcriptome, proteome, and T-DNA mutant libraries [7], as well as using functional analysis of multifunctional genes involved in fungal growth and pathogenicity [8]. Previous studies have revealed that *V. dahliae* contains code for around 530 transcription factors classified into 42 families that play an essential role in host–pathogen interactions [9]. For example, a *V. dahliae* T-DNA mutant lost pathogenicity in cotton when the expression of a flanking gene encoding cytochrome P450 monooxygenase (P450, *VdCYP1*) was considerably suppressed [10]. Pathogenicity has also been linked to melanin production, which is regulated by the cluster-specific transcription factor *VdCmr1* and two other genes within the cluster encoding a polyketide synthase (*VdPKS1*) and a laccase (*VdLac1*) [11]. In a study, using a vector-free split-marker recombination method, the *ExoPG* gene was deleted in *V. dahliae*. The results showed that *ExoPG* was expressed more in the highly aggressive isolate than in the weakly aggressive isolate in response to potato leaf and stem extracts [12]. The function of the isochorismatase hydrolase (ICSH1) gene in *V. dahlia*’s pathogenicity was investigated by Zhu et al. [13], who identified that *VdICSH1* was up-regulated in *V. dahliae* after induction with extracts from potato tissues. Furthermore, Zhu et al. [14] compared in vitro expression of NADPH oxidase (*NoxA*) in highly and weakly aggressive isolates of *V. dahliae* after elicitation with extracts from different potato tissues and found that *NoxA* expression was induced more in the weakly aggressive isolate than in the highly aggressive isolate in response to leaf and stem extracts. 

Fungi produce hundreds of low-molecular-weight and structurally diverse compounds called secondary metabolites or natural products (SMs and NPs, respectively) that function as toxins, antibiotics, pigments, and more [15]. SMs can be categorized into three groups: polyketides derived from acyl-CoAs, terpenes produced from acyl-CoAs, and small peptides derived from amino acids [16]. SMs such as non-ribosomal peptides, terpenes [17], alkaloids, phenols, and polyketides [18], although not considered essential for fungal growth or reproduction [19], have a wide range of functions [20]. For example, SMs are known to play a crucial role in the infection process and disease establishment [21] of numerous pathogenic fungi and in enhancing competitiveness against other microbes [22]. Ascomycete genomes contain an average of 10 non-ribosomal protein synthases (NRPS), 16 polyketide synthases (PKS), and four tryptophan synthetases (TS) and dimethylallyl tryptophan synthetases (DMATS), which is more than the secondary metabolism-related genes identified in basidiomycetes, archeo-ascomycetes, chytridiomycetes, hemi-ascomycetes, and zygomycetes [23]. Due to their high biological activity, SMs are widely used in pharmaceutical research and industry [24,25], i.e., penicillin (a β-lactam antibiotic), lovastatin (a cholesterol-lowering drug) [26], many other antibacterial compounds, and even anti-cancer drugs [27].

Advances in genomics have helped us understand fungal SMs and their biosynthesis at the whole genome scale [28]. For example, genes involved in the biosynthesis of SMs are clustered together, forming what is known as an SM cluster [29]. Moreover, the complete genome sequence of *Penicillium expansum* provided new insights into secondary metabolism biosynthetic gene clusters in fungi, especially those responsible for patulin and citrinin biosynthesis [30]. Genes in clusters are not constitutively expressed, and formerly actively expressed genes can become transcriptionally inactive with repeated culturing [31]. In addition, antiSMASH (a freely available online tool to investigate the presence and diversity of secondary metabolite biosynthesis gene clusters) analyses of fungal genomes showed that ascomycete biotrophs show a lower rate of SM and CAZyme-related gene gain and loss events in the genome [32]. Taylor et al. [33] predicted SM biosynthetic gene clusters in the genome of *Sclerotinia sclerotiorum* and analyzed their gene expression during the infection of *Brassica napus* and reported 80 putative SM clusters. They also found that clusters are homologous to those in the closely related plant pathogen Botrytis cinerea to produce carotenoids, hydroxamate siderophores, dihydroxynaphthalene melanin, and botcinic acid [29]. 

Polyketide-derived metabolites have accounted for most fungal SMs discovered thus far [34]. Polyketides have a variety of structures, ranging from single aromatic-ring compounds such as orsellinic acid, to multi-ring compounds such as aflatoxins and linear structures with an amine group, such as in fumonisin. Polyketides are synthesized from short-chain carboxylic molecules (i.e., acyl-coenzyme A [*CoA*]) in a process such as fatty acid biosynthesis, which is catalyzed by large multi-domain enzymes called polyketide synthases (PKSs) [17]. PKSs in fungi are large, multi-domain enzymes that act iteratively to produce the polyketide chain [35]. Fungal polyketides have remarkable biological activities and include some beneficial molecules used in medicine and agriculture. Lovastatin, for example, is used to lower cholesterol, chaetoviridin is an antibacterial agent, and echinocandins and azoxystrobin are antifungal agents. On the other hand, some fungal polyketides such as aflatoxin, patulin, zearalenone, and fumonisin B1 are harmful mycotoxins that contaminate foods, cost farmers billions of dollars each year, and threaten human, animal and plant health [36].

Recent advances in biosynthetic gene clusters have revealed several fungal *PKSs* that require the action of collaborating enzymes to synthesize the carbon backbone [36]. Fungal PKSs are divided into three groups based on their domain content, activity, and phylogenetic relationships: non-reducing PKSs (NR-PKSs), reducing PKSs (R-PKSs), and partially reducing PKSs (PR-PKSs) [22,37]. NR-PKSs generally synthesize aromatic polyketides and consist of starter unit-specific acyltransferase (SAT), keto synthase (KS), extender unit acyltransferase (MAT), product template (PT), and acyl-carrier protein (ACP) domains. R-PKSs generally synthesize polyketides that consist of a fully saturated carbon chain. R-PKSs contain KS, acyltransferase (AT), dehydratase (DH), methyltransferase (ME), enoyl reductase (ER), keto reductase (KR), and ACP domains. PR-PKSs synthesize aromatic polyketides and share some R-PKS domains such as KS, AT, and KR, the presence or absence of DH and ER domains varies, and they always lack an ME domain. Polyketide products can be modified by other enzymes encoded by genes that cluster together with the PKS gene, such as methyltransferases, oxidoreductases, and transcription factors [38]. Based on their proximity to PKS and similar biosynthetic genes for secondary metabolites, commonly found putative genes can be predicted to be involved in the cluster. The size of polyketide biosynthetic clusters can be small, such as the 18 kb bikaverin cluster in *Fusarium fujikuroi*, or larger, as in the 82 kb aflatoxin biosynthetic cluster in *Aspergillus parasiticus* [39,40].

Gene distance and conserved domains can identify putative loci involved in the prediction of SMs. Furthermore, phylogenetic and comparative genomic analyses are instrumental in identifying gene clusters involved in the production of SMs by making predictions of identical or related compounds that have been characterized in other fungi [41]. 

The main objective of this study was to identify putative PKS genes and their gene clusters in the different species of *Verticillium* genomes and characterize them at the structural and phylogenetic levels by using bioinformatic predictions to identify PKS clusters and the compounds they may produce, with a possible role in pathogenicity.

## 2. Materials and Methods

### 2.1. Full Genome Sequences

Full genome sequences of ten *Verticillium* species were obtained from the National Center for Biotechnology Information (NCBI) (www.ncbi.nlm.nih.gov). The full descriptions and accession numbers of the *Verticillium* genomes examined in this research are displayed in Table 1.

### 2.2. Prediction, Annotation, and Identification of Polyketide Synthase Genes and Gene Clusters

Putative PKS genes and related clusters in the *Verticillium* genomes were predicted using the fungal version of antiSMASH 6.0 known as fungiSMASH at the Secondary Metabolite Bioinformatics Portal (http://www.secondarymetabolites.org [47]) using the FASTA file of the genome sequence assembly as an input. Based on synchronized genome information, antiSMASH predicts which genes might be a part of the biosynthetic cluster. A BLASTp search was conducted against the non-redundant protein sequences in the NCBI database for each PKS or hybrid polyketide synthase-non-ribosomal peptide synthetase (PKS-NRPS) gene identified in *Verticillium* genomes. Furthermore, manual annotation in the CLC genomics workbench version 22 (CLCBio, Aarhus, Denmark) was performed to confirm the borders of each identified PKS cluster. For this purpose, the PKS-related ketosynthase (KS) domain in the full genome sequences of *Verticillium* species was searched using BLASTn as well as BLASTx searches against the whole genome in the CLC genomics workbench. *Verticillium* genomes with sequences of genes previously described to be needed for the PKS biosynthesis cluster were also enquired about. These included transporters, dehydrogenases, transcription factors, cytochrome oxidases, acyltransferases, as well as oxidoreductases. The regions 30 kb upstream and downstream of the predicted PKS genes were then annotated using the WebAUGUSTUS Service (http://bioinf.uni-greifswald.de/webaugustus/ [48]) with the default program parameters based on the *V. longisporum* gene model as the closest available genome to *Verticillium*. The results from both manual and fungiSMASH analyses were evaluated to obtain conclusions about the presence and absence of PKS gene clusters in the *Verticillium* genomes. The conservation of predicted PKS clusters among different *Verticillium* species was evaluated according to the PKS enzyme conservation with BLASTp on identified protein databases (identity > 50% and query coverage > 50%).

### 2.3. Prediction of Verticillium PKS Domains

InterPro Scan tool (https://www.ebi.ac.uk/interpro/ [49]), MOTIF search (http://www.genome.jp/tools/motif/) and NCBI’s Conserved Domain Database [50] were used to identify the domain architecture of each PKS or hybrid PKS-NRPS protein sequence. In addition, the core genes present in the identified clusters were examined for the presence of PKS domains representative of the PKS biosynthesis enzymes using the online PKS-NRPS analysis tool (http://nrps.igs.umaryland.edu/nrps [51]). A putative gene was confirmed as PKS if it contained at least the necessary PKS domains (i.e., KS, AT and ACP) [52]. The results of this analysis were also used to predict the PKS module architecture. Moreover, the PKS genes were categorized based on the presence or absence of reducing (i.e., KR, DH or ER domains) or non-reducing (i.e., SAT, TE, PT and MET) domains.

### 2.4. PKS Cluster Comparison Analysis of Verticillium and Other Fungi

The recovered contigs assumed to be PKS clusters were aligned with sequences of known PKS clusters publicly accessible from other annotated genomes of filamentous fungi. Preservation of synteny of clusters between *Verticillium* genomes and other known clusters from other fungal species investigated by BlastP search in GenBank (E-value cutoff 1e-5, query coverage > 50%, and identity > 35%) and affirmed by manual verification of potential homologous clusters and flanking sequences. *Verticillium* PKS clusters were considered homologous to other known fungal PKS clusters when they shared more than 50% sequence similarity, conjointly when their domain architecture showed high resemblance.

### 2.5. Phylogenetic Analysis

Evolutionary analyses were conducted in MEGA X [53]. For this purpose, the amino acid sequences of predicted PKS genes (in terms of PKS-III) or their KS domains alone (For NR-PKS-I and R-PKS-I sequences) were used to construct phylogenetic trees. Blastp with the *Verticillium* sequences was used to enquiry JGI via MycoCosm [54] as well as NCBI’s protein database. The top blast hit protein sequences identified with an E value ≤ 10^−5^ were used in the analysis. In this analysis, 145 *Verticillium* PKSs (116 R-PKSs, 19 NR-PKSs and 10 PKS-III) and 157 PKSs from other Ascomycetes (128 R-PKSs, 21 NR-PKSs and 8 PKS-III) were used in total. The sequences were then subjected to an online version of MAFFT (multiple alignments using fast Fourier transform) [55] using the iterative refinement method of E-INS-I for alignment which is normally used for sequences with multiple conserved domains. Aligned sequences were observed in BioEdit software (http://www.mbio.ncsu.edu/BioEdit/bioedit.html; accessed on 12 March 2022) and manually trimmed by removing non-aligned sequences. In each case, the best evolutionary model was anticipated using ProtTest [56]. The *Verticillium* PKS evolutionary history was constructed using the maximum likelihood as well as neighbor joining methods with 1000 bootstrap replicates. In all cases, the tree with the highest log-likelihood was plotted (Figure 1 and Figure 2, and Appendix A). The initial tree(s) for the heuristic search were obtained automatically by applying neighbor-joining and BioNJ algorithms to a matrix of pairwise distances estimated using the appropriate models, and then selecting the topology with a superior log-likelihood value. All positions with less than 95% site coverage were eliminated, i.e., fewer than 5% alignment gaps, missing data, and ambiguous bases were allowed at any position (partial deletion option).

## 3. Results

### 3.1. Prediction of Verticillium PKS Domains

Discovering domain architecture in each predicted PKS gene would help determine if all essential domains involved in the functionality of PKS genes are present or not. It would also help to understand whether the final PKS products are reduced or non-reduced. Domain analysis was performed for all the anticipated PKS and PKS-NRPS protein sequences using the PKS-NRPS webpage and motif finder. Since VNR-PKS-I-1 to VNR-PKS-I-4 did not comprise reducing domains, they are most likely non-reducing PKS enzymes. On the other hand, based on all optional domains expected to be present in all highly reducing PKS enzymes, including KR, DH, and ER domains, VRPKS-I-1 to VRPKS-I-26 are likely to be of the reducing type of PKSs enzymes (Figure 3, Table 2 and Table 3). The only exception was VRPKS-I-10, with a domain organization of KS-AT-KR-ACP that was found only in *V. nubilum*. Based on the domain architecture of VRPKS-I-10, this enzyme is likely to be a partially reducing PKS, as it comprises the KR domain but not the complete DH or ER domains. In the VRPKS-I-5 clade, the PKS enzyme in *V. albo-atrum*, *V. zaregamsianum*, *V. isaacii*, and *V. klebahnii* comprised of a KS-AT-DH-MET-ER-KR-C-A-ACP domain organization. The VRPKS-I-5 enzyme in *V. tricorpus* showed an extra NAD_binding_4 domain at the end of the protein sequence. The same unusual domain (NAD_binding_4) was also observed in VRPKS-I-12, where PKS enzymes in *V. isaacii*, *V. klebahnii* and *V. tricorpus* contained a KS-AT-DH-MET-ER-KR-ACP-C-A-ACP-NAD_binding_4 domain architecture, whereas the NAD_binding_4 domain was missing in the *V. zaregamsianum* and *Verticillium albo-atrum* VRPKS-I-12 enzyme. Genomic analyses on the PKS genes and gene clusters of *Verticillium* species revealed that, in total, a minimum of 10 (*V. nonalfalfae* and *V. dahliae*) and a maximum of 19 (*V. klebahnii*) PKSs or hybrid PKS-NRPS biosynthetic gene clusters were present in the genomes of different *Verticillium* species.

### 3.2. Comparison of Verticillium PKS Cluster with Those of Other Fungi

According to the phylogenetic analyses, 19 PKS core enzymes (VRPKS-I-1, VRPKS-I-2, VRPKS-I-3, VRPKS-I-6, VRPKS-I-7, VRPKS-I-8, VRPKS-I-13, VRPKS-I-16, VRPKS-I-17, VRPKS-I-21, VRPKS-I-22, VRPKS-I-23, VRPKS-I-24, VRPKS-I-25, VRPKS-I-26, VNR-PKS-I-1, VNR-PKS-I-2, VNR-PKS-I-3, and VNR-PKS-I-4), showed high resemblance to PKS enzymes from previously known gene clusters. Therefore, comparative genomic approaches were used to predict the probable final products of these secondary metabolite biosynthetic clusters. To this end, flank genes were checked, in addition to the core genes of identified *Verticillium* clusters, and compared to the previously known clusters from other fungal species. These comparisons revealed five interesting PKS-related clusters that displayed previously characterized orthologous gene clusters in other Ascomycetes putatively associated with the biosynthesis of a cytochalasan-like compound (VRPKS-I-3), a fusarielien-like compound (VRPKS-I-7), a fujikurin-like compound (VRPKS-I-8), a melanin compound (VNR-PKS-I-1), and a lijiquinone-like compound (VNR-PKS-I-4). A detailed explanation of the genes proposed for each of the identified *Verticillium* PKS clusters is shown in Appendix A.

#### 3.2.1. Clusters Common among All Examined *Verticillium* Genomes

The VNR-PKS-I-1 gene and cluster highly resembled those from other fungi that produce dihydroxynaphthalene (DHN) melanin (Figure 4A,B and Appendix A). A 1, 3, 8-trihydroxynaphthalene reductase and a cmr1 transcription factor were contained within the *Verticillium* VNR-PKS-I-1 cluster. Homologs of SCD1 and THR1 reductase coding genes were also found in all *Verticillium* species at different locations rather than the contigs in which the initial PKS cluster was found. Interestingly, the melanin biosynthesis cluster in *V. albo-atrum*, *V. isaacii*, *V. zaregamsianum*, *V. klebahnii* and *V. tricorpus* was found next to VRPKS-I-12, which is a PKS-NRPs cluster. Remarkably, the whole VRPKS-I-12 cluster was missing in other *Verticillium* species (Figure 4B, Appendix A).

Orthologs for the PKS-III backbone gene were found in all examined *Verticillium* genomes. A highly conserved single copy of this cluster was also observed in all the *Verticillium* genomes (Figure 5). This cluster is composed of an oxidoreductase, a feruloyl esterase, a fungal-specific transcription factor, a BFR2 protein, a PKS-III, a fructose bisphosphate aldolase, a specific exonuclease, a mitochondrial two oxoglutarate carrier protein, a hypothetical protein, a splicing factor spf30, an iron–sulphur protein, a cell division control protein, and a short-chain dehydrogenase. This cluster thus has all the required gene inventories of a functional PKS cluster. The genes in this cluster exhibited high similarity to the orthologues present in a range of ascomycetes, including *Colletotrichum higginsianum*, *Colletotrichum incanum*, and *Sodiomyces alkalinus* (Figure 5, Appendix A).

Notably, clusters VRPKS-I-17, VRPKS-I-19, VRPKS-I-23, and VRPKS-I-26 are also found in all *Verticillium* genomes. However, they did not show any similarities in terms of gene content to the known PKS clusters from other fungi. 

#### 3.2.2. Clusters Missing in a Few Genomes but Present in Other Species

The VRPKS-I-8 cluster was found only in the pathogenic species, i.e., *V. dahliae* and *V. longisporum*, and displayed a high resemblance to the fujikurin synthase gene (*FfuPks19*) of *F. fujikuroi* and showed significant similarities (39–85% protein identity) to genes in the fujikurin biosynthesis cluster in protein–protein comparisons (Figure 6, Appendix A). The FfuPks19 showed a domain organization of KS-AT-DH-ER-KR. The VRPKS-I-8 has a similar domain organization (Table 2). The fujikurin biosynthesis cluster contains five genes, including a highly reducing PKS (CCT72377), an enoyl reductase domain-containing protein (CCT72378), a hydrolase domain-containing protein (CCT72379), a putative transcription factor (CCT72381), and a putative transporter (CCT72380). Nevertheless, homologs genes of those found in the *F. fujikuroi* fujikurin biosynthesis cluster were found in *V. dahlia* and *V. longisporum* (Figure 6). Hydrolase, transcription factor, and transporter genes were found on different contigs of the *V. longisporum* genome. This could be due to assembly problems. In this case, the sequences were assembled to the reference genome to obtain the whole cluster in one contig. Interestingly, two more genes designated as cytochrome P450 and MFS transporter were found in the VRPKS-I-8 cluster in *V. dahliae* and *V. longisporum*, which were not identified in the FfuPks19 cluster (Figure 6).

The VRPKS-I-20 was found only in *V. isaacii*, *V. klebahnii*, and *V. tricorpus* and showed high protein sequence similarity to an unknown PKS identified in *Penicillium arizonense* (XP_022486633). The VRPKS-I-20 protein exhibited the same domain architecture (KS-AT-DH-MET-ER-KR-ACP) as PKS identified in *Penicillium arizonense.* Synteny comparison between VRPKS-I-20 locus in *V. isaacii*, *V. klebahnii* and *V. tricorpus* genomes and *Penicillium arizonense* reveals extensive conservation (Figure 7). Besides PKS, the cluster comprised of some putative cluster genes, including a mannan endo-1,4-beta-mannosidase coding gene displaying 82% identity with the orthologue present in *Plectosphaerella plurivora* (KAH6695781), a cytochrome P450 exhibiting 75 % identity with the orthologue present in *Colletotrichum fioriniae* (EXF83899), a hypothetical protein exhibiting 72% identity with the orthologue present in *Penicillium arizonense* (XP_022486531), another cytochrome P450 displaying 65% identity with the orthologue present in *Penicillium arizonense* (XP_022486670), and an oxidoreductase protein exhibiting 73% identity with the orthologue present in *Talaromyces rugulosus* (XP_035348268).

The VRPKS-I-6 core gene clustered with burnettramic acid biosynthesis PKS from *Aspergillus burnettii* and other Aspergillus PKS genes, including *A. avenaceus* hypothetical protein (KAE8150730) and *A. alliaceus* burnettramic acids biosynthesis cluster protein A (KAE8392342) (Figure 1). The genes flank to the core gene also showed high similarity in protein sequences to the *A. alliaceus* cluster (Figure 8, Appendix A). VRPKS-I-6 cluster found in *V. zaregamsianum*, *V. isaacii*, *V. klebahnii* and *V. tricorpus* comprised of a PKS-NRPS displaying 67% identity with the orthologue present in *A. avenaceus* (KAE8150730), a hypothetical protein is exhibiting 64% identity with the orthologue present in Didymosphaeria enalia (KAF2268082), a hydrolase exhibiting 58% identity with the orthologue present in *Clohesyomyces aquaticus* (ORY13587), a cytochrome P450 oxygenase displaying 66% identity with the orthologue present in *A. avenaceus* (KAE8150736), a dehydrogenase displaying 77% identity with the orthologue present in *A. avenaceus* (KAE8150737), and another cytochrome P450 oxygenase exhibiting 65% identity with the orthologue found in *A. avenaceus* (KAE8150739). VRPKS-I-6 (KS-AT-DH-ER-KR-ACP-C-A-ACP) exhibited the same domain organization as that of burnettramic acid biosynthesis PKS from *A. burnettii* and *A. avenaceus* (KS-AT-DH-MET-ER-KR-ACP-C-A-ACP) except that it lacks an MT domain (Table 3). Hence, methyl groups may be missing from the final product of the VRPKS-I-6 gene cluster.

VRPKS-I-21 was found in *V. zaregamsianum*, *V. klebahnii* and *V. nubilum* and grouped in a clade with a good bootstrap value to the alternapyrone biosynthesis gene alt5 protein sequence from *Alternaria alternata* and lovastatin diketide synthase *LovF 2* from *Colletotrichum chlorophyte* (Figure 1). The *A. alternata Alt5* protein showed the following domain organization: KS-AT-DH-MT-ER-KR-ACP. The VRPKS-I-21 showed the same domain architecture, except it lacks an ACP domain. In the VRPKS-I-21 biosynthetic cluster, VRPKS-I-21 is sitting next to some putative cluster-related genes such as NADH-ubiquinone oxidoreductase, cytochrome p450 domain-containing protein, and drug resistance protein, but not the oxidase or the other two cytochrome P450 proteins that may be part of the alternapyrone biosynthesis gene cluster (Appendix A).

VRPKS-I-22 was found only in pathogenic *V. dahliae*, *V. longisporum*, *V. alfalfae*, and *V. nonalfalfae* and showed a high resemblance in protein sequence to the fusaric acid gene cluster core gene, PKS6 (Figure 1, Appendix A). VRPKS-I-22 exhibited the same domain organization (KS-AT-DH-MET-ER-KR-ACP) as that of fusaric acid biosynthesis PKS from *F. mexicanum* (KAF5539484) and *F. miscanthi* (ALQ32877). However, the rest of the genes in the VRPKS-I-22 locus did not show any similarities to the genes needed for fusaric acid production. The VRPKS-I-22 cluster is comprised of a sulfate permease, displaying 51% identity with the orthologue present in *Trematosphaeria pertusa* (XP_033684343); a hydrolase, exhibiting 62% identity with the orthologue present in *Monosporascus* sp. (RYP55738); two hypothetical proteins, exhibiting 60% and 79% identity with the orthologues present in *Fusarium graminearum* (CZS74879) and *Colletotrichum musicola* (KAF6840530), respectively; an MSF-1 protein, exhibiting 85% identity with the orthologue present in *Sodiomyces alkalinus* (XP_028463299); a methionine aminipeptidase, displaying 82% identity with the orthologue present in *Colletotrichum sidae* (TEA17142); a peroxiredoxin family protein, displaying 67% identity with the orthologue present in *Sodiomyces alkalinus* (XP_028463297); and an amino acid aminotransferase, exhibiting 75% identity with the orthologue present in *Sodiomyces alkalinus* (XP_028463295). On the other hand, the five putative fusaric acid cluster genes encode proteins with resemblance to a homoserine-*O*-acyltransferase, a hypothetical protein, a serine hydrolase, an aspartate kinase, and a polyketide synthase (Appendix A).

#### 3.2.3. Species-Specific Clusters

The VNR-PKS-I-4 gene cluster was found only in *V. nubilum*. Protein sequences of *V. nubilum* VNR-PKS-I-4 genes revealed a ~48 kb cluster, comprised of 12 open reading frames with close resemblance to the lijiquinone biosynthesis cluster found in *Muyocopron atromaculans* (Figure 9, Appendix A). The protein sequences of the *V. nubilum* VNR-PKS-I-4 genes showed high similarity to putative hydroxylase exhibiting 78−81% identity with the orthologue present in *M. atromaculans* (XP_007816049) and *Dactylonectria macrodidyma* (KAH7133940), a hypothetical protein exhibiting 49% identity with the orthologue present in Hypoxylon sp (OTB08973), an R-PKS (VRPKS-I-15) displaying 61−68% identity with the orthologue present in *Dactylonectria macrodidyma* (KAH7133936) and *Fusarium fujikuroi* (KLP04343), a short chain dehydrogenase exhibiting 66% identity with the orthologue present in *Ilyonectria destructans* (KAH7019726), a serine hydrolase exhibiting 54−61% identity with the orthologue present in *M. atromaculans* (PVH94008) and *Dactylonectria macrodidyma* (KAH7133931), a putative short-chain dehydrogenase exhibiting 74% identity with the orthologue found in *M. atromaculans* (XP_007816049), another putative hydrolase displaying 74% identity with the orthologue present in *Dactylonectria macrodidyma* (KAH7133934), a putative oxidase exhibiting 71% identity with the orthologue present in *M. atromaculans* (KEQ59244), an NR-PKS (VNR-PKS-I-4) exhibiting 68−61% identity with the orthologue present in *M. atromaculans* (AMJ52084), *Pseudogymnoascus destructans* (XP_024319559) and *Dactylonectria macrodidyma* (KAH7133936), an inorganic phosphate transport displaying 42% identity with the orthologue present in *Scedosporium apiospermum* (XP_016645996), a putative O-methyltransferase exhibiting 86% identity with the orthologue present in *Dactylonectria macrodidyma* (KAH7133929) and a putative ariadne-like RING finger protein R811 exhibiting 56% identity with the orthologue present in *Colletotrichum siamense* (XP_024319559) (Appendix A). The *M. atromaculans* R-PKS found in the lijiquinone biosynthesis cluster showed a KS-AT-DH-MT-ER-KR-ACP domain organization. *V. nubilum* VRPKS-I-15 showed the same domain architecture but lacked an ACP domain. The domain organization of the second PKS in the cluster, VNR-PKS-I-4 (KS-AT-ACP-MT-R) and lijiquinone biosynthesis NR-PKS (SAT-KS-AT-PT-ACP-MET-R) was somewhat different, as the SAT and PT domains were missing in VNR-PKS-I-4.

VRPKS-I-3, a PKS-NRPS cluster, was found only in the *V. albo-atrum* genome and grouped in a clade with a good bootstrap value to the cytochalasans biosynthesis PKS sequences, with the most significant similarity to pyrichalasin H synthase protein from *Pyricularia oryzae* (A0A4P8WAE5) (Figure 1). The fungiSMASH could not predict a product for the *V. albo-atrum* VRPKS-I-3 cluster, but it likely allows for the formation of cytochalasans in this fungus. VRPKS-I-3 had a KS-AT-DH-ER-KR-C-A-ACP domain organization, identical to the domain organization of pyrichalasin H synthase protein. Both *V. albo-atrum* VRPKS-I-3 and *P. oryzae* cytochalasan clusters were predicted to have high similarity in terms of protein sequences and gene order (Figure 10, Appendix A).

VRPKS-I-7 was found only in one copy in *V. zaregamsianum* and grouped in a clade with the fusarielin biosynthesis gene PKS-9 protein sequence from *Fusarium pseudograminearum*. VRPKS-I-7 showed a high protein sequence similarity of 82% with the *F. pseudograminearumas* PKS9 fusarielin biosynthesis (XP 009262583) protein with an identical PKS domains sequence of KS-AT-DH-MET-ER-KR. In addition to a highly reducing PKS gene, the cluster contains a protein with a thioesterase (FSL2) domain (FGSG_10463) exhibiting 61% sequence identity with the orthologue present in *V. zaregamsianum* VRPKS-I-7 cluster, an epimerase (FSL3)-containing domain protein (FGSG_17368) exhibiting 55% identity with the orthologue present in VRPKS-I-7 cluster, a cytochrome P450 oxygenase (FSL4) protein (FGSG_10461) exhibiting 57% identity with the orthologue present in VRPKS-I-7 cluster, an enoyl reductase (FSL5) protein (FGSG_17367) with no homologs found in *V. zaregamsianum* VRPKS-I-7 cluster, and a transcription factor (FSL7) protein (FGSG_10458) exhibiting 85% identity with the orthologue present in VRPKS-I-7 cluster. The fusarielin biosynthesis gene cluster also contains a hypothetical protein (FGSG_10459) with an unknown role in the fusarielin biosynthesis pathway exhibiting 85% identity with the orthologue present in the VRPKS-I-7 cluster (Figure 11, Appendix A).

None of the other *Verticillium* R-PKS clusters showed similarities to previously identified or unknown clusters from other fungi. However, based on their gene content and cluster-related genes such as transporters, cytochrome P-450, hydrolases, dehydrogenases, and transcription factors, some of these *Verticillium* R-PKS clusters such as VRPKS-I-1, VRPKS-I-9, VRPKS-I-17 and VRPKS-I-25 are likely to represent a functional PKS biosynthesis cluster.

### 3.3. Missing PKS Gene Clusters in Closely Related Species

To determine which PKS or PKS-NRPS genes/clusters were present in different *Verticillium* species but absent in closely related species or vice versa, we examined the distribution patterns of these genes. Such distribution patterns were observed in seven PKS or PKS-NRPS genes. For instance, VRPKS-I-2 was found in *V. albo-atrum*, *V. isaacii*, *V. tricorpus* and *V. klebahnii*, but not in *V. zaregamsianum*. The presence of the VRPKS-I-8 gene (likely involved in fujikurine biosynthesis) in *V. dahliae* and *V. longisporum* and its absence in closely related species (*V. alfalfae* and *V. nonalfalfae*) is also in congruence with gene gain/loss events. However, more analysis is needed to confirm this statement. In another example, VRPKS-I-11 was absent in *V. dahliae* but was present in *V. longisporum*, *V. alfalfae* and *V. nonalfalfae*. In *V. dahliae*, the second PKS-NRPS gene along with seven other genes next to VRPKS-I-11 was missing (Figure 12). The VRPKS-I-9 cluster was lost in *V. zaregamsianum*, but intact homologs of this cluster were observed in *V. isaacii*, *V. tricorpus* and *V. klebahnii*. Interestingly, the whole VRPKS-I-9 cluster was also observed in *V. alfalfa* and *V. nonalfalfae*. Homologs of the VRPKS-I-18 cluster were found in *V. isaacii*, *V. zaregamsianum* and *V. klebahnii*, but not in *V. tricorpus*. Such PKS gene gain/loss event was also found in VRPKS-I-21 homologs. The VRPKS-I-21 gene cluster was observed in *V. zaregamsianum* and *V. klebahnii* but was absent in *V. isaacii* and *V. tricorpus*. It is also worth noting that the homolog of the VRPKS-I-21 cluster was found in *V. nubilum*, a distantly related species.

### 3.4. Phylogenetic Analysis

The NR-PKS phylogenetic tree predicted five *Verticillium* clades, labeled as VNR-PKS-I-1 to VNR-PKS-I-4. The NR-PKS phylogenetic analysis confirmed that these predicted VNR-PKSs were present in all *Verticillium* genomes clustered with previously described enzymes. The presence of VNR-PKSs in different *Verticillium* species is illustrated in Figure 2 and Figure 3. The VNR-PKS-I-1 sequences were found in all *Verticillium* genomes and grouped with melanin biosynthesis PKSs from different Colletotrichum species. VNR-PKS-I-2 was only found in *V. zaregamsianum*, and *V. klebahnii* clustered with grayanic acid (GRA) synthase which is involved in the production of orcinol depsidone specific to in *Cladonia grayi* [57]. VNR-PKS-I-3 was detected in the genomes of *V. dahliae*, *V. longisporum*, *V. alfalfae* and *V. nonalfalfae*, forming a distinct and well-supported clade with orsellinic acid synthase, *OpS1* which is responsible for producing oosporein toxin synthesized by entomopathogenic fungus and *Beauveria bassiana* [58]. VNR-PKS-I-4 was found only in *V. nubilum* and clustered with the azaphilone polyketide synthesis coding gene.

Phylogenetic analysis of the KS domain of the predicted R-PKS-I protein sequences separated *Verticillium* species into 26 well-supported clades, labeled from VRPKS-I-1 to VRPKS-I-26. The presence of VRPKSs in different *Verticillium* species is illustrated in Figure 1 and Figure 3. The VRPKS-I-1 clade consists of sequences from *V. dahliae*, *V. zaregamsianum*, *V. alfalfa*, and *V. longisporum*. The VRPKS-I-2 clade included sequences from the closely related species *V. isaacii*, *V. klebahnii*, *V. tricorpus*, and *V. albo-atrum*. The VRPKS-I-3 clade was observed only in the *V. albo-atrum* genome. The VRPKS-I-4, VRPKS-I-10, and VRPKS-I-15 clades contained sequences found only in *V. nubilum*. The VRPKS-I-5, VRPKS-I-12, VRPKS-I-16, and VRPKS-I-25 clades consist of sequences from *V. isaacii*, *V. klebahnii*, *V. tricorpus*, *V. zaregamsianum*, and *V. albo-atrum*. The VRPKS-I-6, VRPKS-I-13, and VRPKS-I-14 clades included KS sequences from *V. isaacii*, *V. klebahnii*, and *V. tricorpus*, *V. zaregamsianum*. The VRPKS-I-7 clade was observed only in the *V. zaregamsianum* genome, while the VRPKS-I-8 clade was detected only in the *V. dahliae* and *V. longisporum* genomes. The VRPKS-I-9 clade included sequences from *V. isaacii*, *V. klebahnii*, *V. tricorpus*, *V. albo-atrum*, *V. alfalfae*, and *V. nonalfalfae*. The VRPKS-I-11 and VRPKS-I-22 clades were comprised of KS sequences from *V. dahliae*, *V. nonalfalfae*, *V. alfalfae*, and *V. longisporum*. The VRPKS-I-17, VRPKS-I-19, VRPKS-I-23, and VRPKS-I-26 clades were found in all *Verticillium* genomes. The VRPKS-I-18 clade consists of KS sequences from *V. isaacii*, *V. klebahnii*, and *V. zaregamsianum*. The VRPKS-I-20 clade included sequences from *V. isaacii*, *V. klebahnii*, and *V. tricorpus*. Finally, the VRPKS-I-21 clade contained KS sequences from *V. klebahnii*, *V. zaregamsianum*, and *V. nubilum*. 

The HR-PKS phylogenetic analysis showed that 15 out of the 26 predicted *Verticillium* HR-PKSs (VRPKS-I-1, VRPKS-I-2, VRPKS-I-3, VRPKS-I-6, VRPKS-I-7, VRPKS-I-8, VRPKS-I-13, VRPKS-I-16, VRPKS-I-17, VRPKS-I-21, VRPKS-I-22, VRPKS-I-23, VRPKS-I-24, VRPKS-I-25, and VRPKS-I-26) grouped with previously described enzymes. VRPKS-I-1 and VRPKS-I-2 grouped with oxaleimides biosynthesis-related genes, poxE from *Penicillium oxalicum*, and fusaridione A synthase and fsds from Fusarium heterosporum. VRPKS-I-3 clustered with the clade that contains ffsA, which is involved in cytochalasin production. VRPKS-I-6 was grouped in a clade that contains burnettramic acids biosynthesis cluster protein-coding A. VRPKS-I-7 and VRPKS-I-8 were grouped with fusarielin toxin biosynthesis gene PKS-9 from *Fusarium pseudograminearum* and fujikurin synthase coding gene from *F. fujikuroi*, respectively. VRPKS-I-13 was clustered with sdnO, involved in sordarin biosynthesis, a glycoside antibiotic with the ability of protein inhibition in fungi produced by *Sordaria araneosa* [59] and an unknown PKS gene from *Penicillium* sp. VRPKS-I-16 and VRPKS-I-17 were grouped with the T-toxin synthase PKS2 from *Cochliobolus heterostrophus* and the reducing polyketide synthase from Colletotrichum species which is also involved in T-toxin biosynthesis [60]. VRPKS-I-21 clustered in a clade containing the *alt5* PKS gene. VRPKS-I-22 and VRPKS-I-23 clustered with fusaric acid synthase coding gene FUB1 from Fusarium species and phenolpthiocerol synthesis polyketide synthase *ppsA* from *Sodiomyces alkalinus*, respectively [61,62]. VRPKS-I-23 was clustered with *FUB1* from the Colletotrichum species. VRPKS-I-24 was found only in *V. albo-atrum* and grouped with a PKS gene involved in trichothecene biosynthesis. VRPKS-I-25 and VRPKS-I-26 grouped with fungal maleidrides, scytalidin scyPKS from *Stachybotrys chartarum*, and fumagillin synthase from *Aspergillus fumigatus* [63]. VRPKS-I-26 also clustered with the phaseopelide biosynthesis coding gene *amplA*, a fungal aliphatic macrolide from *Arthrinium phaeospermum* [64]. The remaining R-PKS genes from *Verticillium* species did not show any resemblance to known PKS cluster-related genes. Hence, 15 out of 26 R-PKS genes were grouped with previously known PKS genes from other Ascomycetes, allowing us to further investigate their putative functions. *Verticillium* R-PKS-I phylogeny analysis proposed independent and multiple evolutionary origins for the PKS gene in the *Verticillium* genomes. 

## 4. Discussion

### 4.1. PKS Identification

Although polyketides have been claimed to be pathogenicity factors in fungi, our knowledge of PKSs in different members of *Verticillium* is very limited. In this study, using ten publicly available *Verticillium* genome sequences, we found that different species of *Verticillium* differ in PKS-related genes/gene clusters and possible secondary metabolite production. For example, *V. klebahnii* and *V. tricorpus* showed much higher PKS clusters than *V. alfalfae* and *V. dahliae*. The genome analysis of *V. dahliae* has already revealed secondary metabolite genes and gene clusters [21]. Similar research was conducted by Hansen et al. [65], in which they compared PKSs and non-ribosomal peptide synthetases (NRPSs) from ten different Fusarium species and found 52 NRPS and 52 PKS orthology groups.

In this study, we identified a minimum of 10 (*V. nonalfalfae* and *V. dahliae*) and a maximum of 19 PKS or hybrid PKS-NRPS biosynthetic gene clusters in the *Verticillium* genomes and predicted products synthesized by these gene clusters. All the *Verticillium* PKS and hybrid PKS-NRPS genes encode enzymes that include all the required PKS domains. With each *Verticillium* PKS or hybrid PKS-NRPS, we were able to determine which genes would be part of biosynthetic clusters. Besides phylogenetic analysis, we also searched for homologs of the identified clusters in other fungi. Based on phylogenetic and genomic data and chemical structure considerations, we propose that five PKSs in the VNR-PKS-I-1, VNR-PKS-I-4, VRPKS-I-3, VRPKS-I-7, and VRPKS-I-8 clades are involved in synthesizing known products that include melanin, lijiquinone, cytochalasin, fusarielin and fujikurin, respectively. 

We showed that most PKS genes found in *Verticillium* species are in predicted SM biosynthetic clusters, suggesting that the increase in PKS gene copy number specifically involves genes related to secondary metabolite production. Fatema et al. [66] reported similar findings in *C. rosea* and stated that most PKS genes were in predicted SM biosynthetic clusters. In the present study, the PKSs were divided into three (VNR-PKSs, VRPKSs and PKS-III) groups and further divided into 32 well-supported PKS gene clades. Based on the domain structures in each clade, we speculate that PKS homologs within each of the 32 R-PKS clades are responsible for the synthesis of the same polyketide structure that is different from polyketides synthesized by PKSs in other clades. 

Homologs of the PKS clades VRPKS-I-17, VRPKS-I-19, VRPKS-I-23, and VRPKS-I-26 were detected in all *Verticillium* species. The groupings among the sequences in these clades largely resemble the *Verticillium* taxonomic expectations. This finding is consistent with studies conducted by Brown and Proctor [22], who identified 488 PKS genes in 31 Fusarium species and stated multiple and disparate patterns of distribution. The occurrence of the PKS genes in all species has been documented in previous studies [67,68]. Phylogenetic and comparative analyses of PKS homologs using different species of fungi can help find new SMs and understand SM biosynthesis pathways [69]. The discontinuous distribution of SM biosynthetic genes in fungi has been attributed to multiple genetic processes such as gene gain or loss [70] and horizontal gene transfer (HGT) [71] affecting single genes or gene clusters, providing explanations for the diversity and taxonomic distribution of SM gene clusters across fungal species [72].

### 4.2. Clusters Common among All Examined Verticillium Genomes

The VNR-PKS-I-1 sequences were found in all *Verticillium* genomes and grouped with PKSs from different Colletotrichum species that have been associated with DHN melanin biosynthesis [73]. This conserved gene cluster was found in all *Verticillium* genomes examined. Melanin biosynthesis requires *SCD1* and *THR1* reductase. In addition, a 1,3,8-trihydroxynaphthalene reductase and a cmr1 transcription factor are required for DHN fungal melanin biosynthesis [73] and are contained within the *Verticillium* VNR-PKS-I-1 cluster. Shi-Kunne et al. [21] identified 11 potential PKS genes/clusters in the *V. dahliae* genome, including loci that can be implicated in DHN-melanin and fujikurin production. It has been proven that the DHN melanin biosynthesis pathway in other fungi also requires scytalone dehydratase *SCD1* and *THR1* reductase coding genes in order to function [73]. In *V. dahlia,* mutation of either the VNR-PKS-I-1 or transcription factor *VdCmr1*, resulted in the production of transparent transformants that also reduced the responses of *V. dahliae* to stress-related factors [11]. In another study, Li et al. [74] screened the effect of *VdPKS9* (VR-PKS-I-17 in our study) deletions in the virulent strain of *V. dahliae* and found that *VdPKS9* negatively controls the VdPKS1 (VNR-PKS-I-1)-associated melanin pathway.

The analysis has putatively linked type III PKSs found in all *Verticillium* genomes with a similar PKS-III gene in other ascomycetes. This cluster was conserved in all genomes examined in this study and showed high similarity in gene order and sequence with several Colletotrichum strains. Type III PKSs produce SMs with various biological activities, including antimicrobial activity. Despite extensive research in plants and bacteria, only a few type III PKSs from fungi have been identified [75]. Type III PKSs, also known as the chalcone synthase-like PKSs, were initially found in plants and then in bacteria. Later, with progress in genomics analysis, four type III PKS genes (CsyA, CsyB, CsyC, and CsyD) were identified in *Aspergillus oryzae* and type III PKS genes in other fungi [76]. Phylogenetic analyses of type III PKSs from plants, bacteria and fungi showed a unique origin for the fungal clade [77,78]. Sayari et al. [79] examined 20 genomes of Ceratocystidaceae and reported that the PKS III-containing gene cluster is highly conserved and present in all 20 of the genomes examined. PKS III genes encode polyketide pyrones and resorcinols [80].

### 4.3. Species-Specific Clusters

VRPKS-I-3 is clustered with the clade that contains *ffsA*, which plays a role in cytochalasan production. Cytochalasins or cytochalasans are a large family of structurally diverse macromolecular fungal polyketide synthase non-ribosomal peptide synthetase (PKS-NRPS) hybrid secondary metabolites [81]. Cytochalasins produced by *Aspergillus flavipes* and the pyrichalasin H synthase coding gene are important toxins produced by *Pyricularia oryzae* that show substantial cytotoxicity [82]. Cytochalasins, such as cytochalasin E produced by *Aspergillus clavatus*, are polyketide-amino acid hybrid molecules that belong to the cytochalasan family of fungal secondary metabolites [83]. Generally, to form a cytochalasin biosynthesis cluster, four genes are essential, including PKS-NRPS, enoyl reductase, putative Diels−Alderase (*pDA*), and α/β hydrolase (*HYD*) genes [84]. All four of these genes were present in V. albo-atrum VRPKS-I-3 cluster. 

In the phylogenetic analysis, VNR-PKS-I-4, which is found only in the genome of *V. nubilum*, clustered with the azaphilone polyketide synthesis coding gene lijiquinone and Muyocopron and chaetoviridin (caz) biosynthesis-related PKS from *Chaetomium globosum* [85,86]. Based on the cluster comparison analyses, it can be anticipated that the final product of the VNR-PKS-I-4 gene cluster, which was found only in *V. nubilum*, is an antifungal compound in the azaphilone group. This is because the comparison analysis revealed that the genes flank to the VNR-PKS-I-4 gene showed high protein sequence similarity to the previously known azaphilone polyketide lijiquinone cluster identified in Chinese yew fungal endophyte *M. atromaculans* [85].

The VRPKS-I-7 gene cluster, found only in the genome of *V. zaregamsianum*, was putatively linked with the fusarielin biosynthesis cluster of Fusarium pseudograminearuum. Fusarielins are decalin core polyketides produced by several species of Fusarium and Aspergillus. Sieber et al. [87] showed that the genes of the metabolites aurofusarin and fusarielin are conserved in the closely related *F. pseudograminearum* but were absent in other Fusarium species. It has also been proven that the fusarielin biosynthesis cluster of *F. pseudograminearuum* was obtained through HGT. Likewise, in the current analysis, the genes for this cluster were only found in *V. zaregamsianum* and were not present in other genomes examined. Droce et al. [88] have proposed that the fusarielin biosynthesis gene cluster of *F. pseudograminearumas* is composed of seven genes. The comparison analysis showed that six out of seven genes needed for the biosynthesis of fusarielin are also present in the VRPKS-I-7 cluster. The VRPKS-I-7 core gene also matched with *A. avenaceus* (58.44%), *Aspergillus wentii* (58.29%), *Sarocladium implicatum* (78.19%), *Whalleya microplaca* (57.63%), *Penicillium nordicum* (56.06%), and *Penicillium camemberti* (56.15%). Therefore, the VRPKS-I-7 cluster might be gained through HGT from distantly related species, although more investigation is required to prove it. 

### 4.4. Clusters Missing in a Few Genomes but Present in Others

The VPKS-8 gene cluster was putatively linked with the fujikurin biosynthesis cluster identified in *Fusarium fujikuroi*. This cluster is found only in highly pathogenic *Verticillium* species, including *V. dahliae* and *V. longisporum*. *Fujikurins* are natural products isolated from the fungus *F. fujikuroi*. It has been reported that the PKS-19 gene controls fusarielin toxin biosynthesis in *F. pseudograminearum* and the fujikurin synthase coding gene in *F. fujikuroi* [88,89]. *Fujikurins* and fujikurin-like compounds can play important roles in host–pathogen interactions in several pathogens, including some economically significant plant pathogenic fungi [90]. The core gene of the VPKS-8 gene cluster showed a high similarity to the PKS-19 gene from *F. fujikuroi*. Moreover, the fujikurin biosynthesis cluster contains five genes [89]. The homologs of all these five genes were also found in *V. dahliae* and *V. longisporum*. Since this cluster has been found only in highly pathogenic species of *Verticillium*, we hypothesize that the putative fujikurin-like compound gene cluster plays a significant role in pathogen infection and lifecycle, as in *Fusarium*.

### 4.5. Clusters Found in Distantly Related Fungi

Some of the *Verticillium* R-PKS-I clusters found in this study were also found in distantly related fungi. For example, from the phylogenetic analysis results, the VRPKS-I-1 was grouped with oxaleimides biosynthesis-related genes, such as poxE from *P. oxalicum* [91,92]. In another example, VRPKS-I-20 was found only in *V. isaacii*, *V. klebahnii*, and *V. tricorpus* and showed high protein sequence similarity to an unknown PKS identified in *P. arizonense* (XP_022486633) [93]. As mentioned earlier, the VRPKS-I-6 is also grouped in a clade that contains burnettramic acids biosynthesis cluster protein-coding A, an exceptional PKS-NRPS-derived bolaamphiphilic pyrrolizidinedione with antifungal activity from *A. burnettii* and *A. alliaceus* [94]. VRPKS-I-21 also clustered in a clade containing *alt5*, a PKS gene involved in the production of decaketide compound, alternapyrone, from *Alternaria solani* [95].

Large-scale genomic and functional comparisons elucidated the various proposed mechanisms that might explain the evolution of host adaptation in *Verticillium* species, such as enrichment for transposable elements, extensive chromosomal rearrangements, additional dispensable chromosomes, a high proportion of positively selected genes, horizontal gene transfer, and gene gain or loss [96]. Phylogenetically distant or unrelated species can exchange genetic material via horizontal gene transfer (HGT) which could explain why some of the above-mentioned clusters are present in *Verticillium* genomes. The horizontally transferred genes can benefit the recipient organism by boosting its ability to withstand environmental stress, colonize hosts, access nutrients, or gain an evolutionary advantage. However, more research is needed to validate this hypothesis.

### 4.6. Comparison of the Number of PKS Genes in Pathogenic and Non-Pathogenic Verticillium Strains

In this study, we found a greater number of PKS clusters in non-pathogenic/weakly aggressive species of *Verticillium* compared to the pathogenic ones. Pathogens and plants are evolving in a perpetual arms race, resulting in the relatively fast evolution of effectors and resistance (R)- and susceptibility (S)-related proteins. In general, effectors are called virulence factors if they promote virulence. On the contrary, effectors that are recognized by R proteins become avirulence (Avr) factors (Avr proteins). These avirulence and virulence factors, determining resistance and susceptibility of plants, respectively, are identified in many plants’ pathogenic fungi [97,98]. Plants have transferred Ave1 to *Verticillium* through HGT. Ave1 is a secreted protein recognized by Ve1 in tomatoes, and Ave1 expression is induced during host colonization [99]. The findings show that pathogenic *Verticillium* species such as *V. dahliae*, *V. longisporum*, *V. albo-atrum*, *V. alfalfae*, and *V. nonalfalfae* have fewer polyketide clusters than non-pathogenic and weekly aggressive species such as *V. tricorpus*, *V. zaregamsianum*, *V. nubilum*, *V. isaacii*, and *V. klebahnii*. Although it has been reported that the polyketide synthase VdPKS9 plays an important role in melanin biosynthesis and hyphal growth to promote *V. dahlia*’s virulence, it has also been reported that mutation of either the transcription factor *VdCmr1* or the VdPKS1 did not reduce virulence and both genes were not required for pathogenesis on tobacco and lettuce [11]. These results concur with the idea that to maintain their virulence, pathogenic fungi tend to lose genes that might be targeted or recognized by plants.

### 4.7. Missing PKS Gene Clusters in Closely Related Species

When it comes to genetic variation, fungi have long been thought to be limited because they reproduce asexually. Notably, adaptive evolution occurs regardless of the absence of meiotic recombination in asexual organisms. The different mechanisms of adaptive evolution can lead to genetic divergence over long periods, which may eventually lead to the emergence of new species [100]. Multiple evolutionary processes, including horizontal transfer and loss/deletion of the genes, have been proposed to contribute to such discontinuity in *Verticillium* genomes. In the current study, such distribution patterns were observed in seven PKS or PKS-NRPS genes. For instance, VRPKS-I-2 was found in *V. albo-atrum*, *V. isaacii*, *V. tricorpus*, and *V. klebahnii*, but not in *V. zaregamsianum*. On the other hand, *V. zaregamsianum* obtained a VRPKS-I-2 gene cluster which has also been found in distantly related species to *V. zaregamsianum*, including *V. dahliae*, *V. longisporum*, *V. alfalfa*, and *V. nonalfalfae*. The presence of the VRPKS-I-8 gene (likely involved in fujikurine biosynthesis) in *V. dahliae* and *V. longisporum* and its absence in closely related species (*V. alfalfae* and *V. nonalfalfae*) is also in congruence with gene gain/loss events. Genome sequencing projects have revealed large and frequent changes between species in the size of gene families. Numerous morphological, physiological, behavioral, and genetic differences between species can be traced back to these changes. They are also thought to be responsible for much of the natural genetic and genomic diversity that exists. Gene duplications and losses have also been found to contribute to long-term differences in the size of gene families between species [101]. Gene loss was frequently overlooked as an evolutionary driver in the past, mainly because it was associated with the elimination of redundant gene duplicates that had no obvious functional consequences [102]. Recent genomic data, on the other hand, suggest that gene loss may be a major cause of phenotypic diversity and fungal habitat change [103]. For example, gains, losses, or changes in the function of TRI genes have contributed to the structural diversity of trichothecenes, a family of toxic secondary metabolites produced by various Aspergillus, Fusarium, Trichoderma, and Trichothecium fungal species [104]. Comparative genomic analysis of *V. dahliae* strains revealed extensive large-scale genomic rearrangements, likely facilitated by incorrect double-stranded break repair, which led to lineage-specific genomic regions that are enriched for planta-expressed effector genes encoding secreted proteins that aid in host colonization [105,106].

### 4.8. PKS-NRPS Cluster Comparison among Different Verticillium Strains

PKS-NRPSs are formed because of the fusion of PKS and NRPS domains and are considered among the most important groups of clusters due to their structural complexity [107]. In this study, a total of nine PKS-NRPS genes were found. The VRPKS-I-1 cluster was found in *V. dahliae*, *V. alfalfae*, *V. longisporum*, and *V. zaregamsianum*. The analyses could not link this cluster to any previously known cluster. However, it has been proven that the nag 1 gene identified in this cluster is involved in the pathogenicity of *V. dahliae* [108]. Functional analysis of nag 1 by dsRNA-mediated gene silencing showed that the lack of nag 1 exponentially reduced conidial production and fungal growth. In addition, the Nag1-silenced mutants displayed noticeable deficiencies in fungal virulence and microsclerotia formation. In alignment with a decrease in microsclerotia formation, melanin biosynthesis and transcripts of genes involved in melanin formation were significantly decreased in Nag1-silenced mutants. Hence, although more functional analysis is needed, it seems that the VRPKS-I-1 cluster plays the same roles in *V. alfalfae*, *V. longisporum*, and *V. zaregamsianum*. Interestingly, when we searched for the nag 1 gene in these clusters, we could find it in the VRPKS-I-1 cluster of *V. dahliae*, *V. alfalfae*, and *V. longisporum*. However, in *V. zaregamsianum*, this gene was found in a different contig, and it seems that it is not part of the same cluster.

## 5. Conclusions

The genome comparison approach has provided a significant understanding of fungal pathogens’ evolution. In this study, we used a combination of phylogenetic and genomic approaches to identify 32 putative PKS genes and possible clusters in the genomes of *Verticillium* members. All the identified clusters seem to be complete and hence, possibly functional. We anticipate that at least five clusters, including those for putative DHN-melanin-, cytochalasin-, fusarielien-, fujikurin-, and lijiquinone-like compounds, may belong to the active PKS repertoire of *Verticillium*. Further functional analyses will shed light on the role of the identified clusters. Our findings provide a solid framework for further research into the functionality of polyketide biosynthesis gene clusters in this commercially significant fungal genus.

## Figures and Tables

**Figure 1 biology-11-01252-f001:**
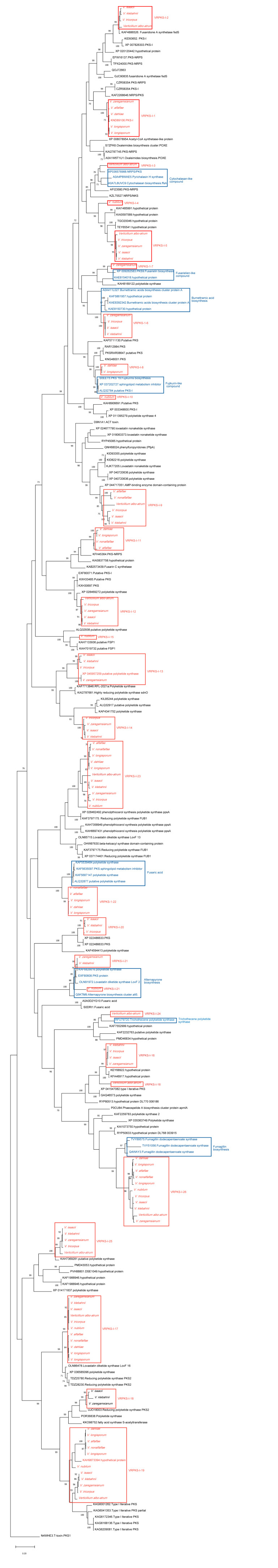
Maximum likelihood tree of 116 and 128 *Verticillium* and other fungi beta-ketoacyl synthase (KS) domains of the reducing PKS-I gene sequences, respectively, that were used in this study. The 26 main clades recovered for the *Verticillium* species are labeled from VRPKS-I-1 to VRPKS-I-26. Similar groups were obtained with the neighbor-joining analysis. GenBank accession numbers for each of these sequences are provided. In this analysis, we used 1000 bootstrap repeats as indicated in the internodes.

**Figure 2 biology-11-01252-f002:**
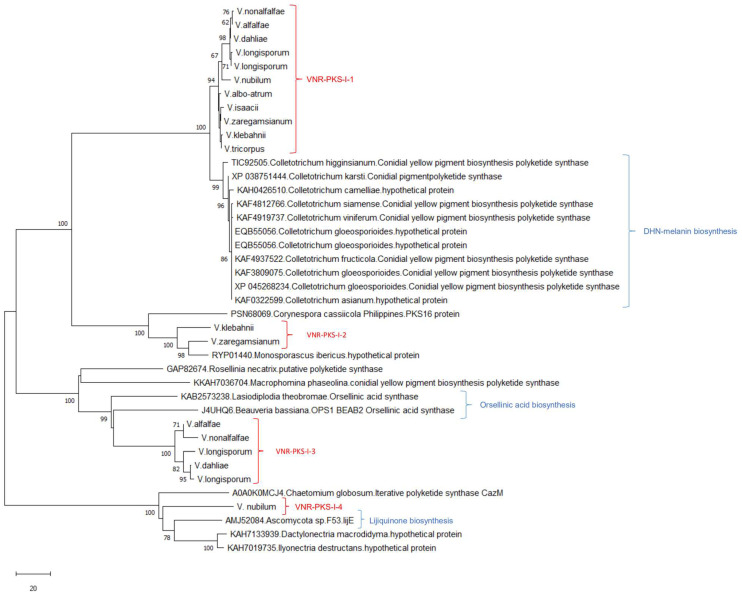
Neighbor-joining tree inferred from nucleotide sequences of non-reducing PKS coding regions of *Verticillium* species as well as NR-PKS sequences obtained from other Ascomycetes. GenBank accession numbers for each sequence are provided. Similar groups were obtained with the maximum likelihood analysis. In this analysis, we used 1000 bootstrap repeats as indicated in the internodes.

**Figure 3 biology-11-01252-f003:**
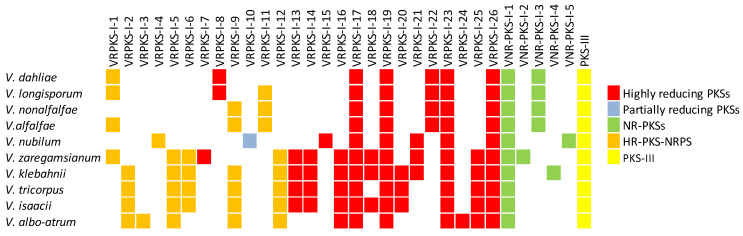
Predicted PKS and PKS-NRPS biosynthetic genes in different *Verticillium* species.

**Figure 4 biology-11-01252-f004:**
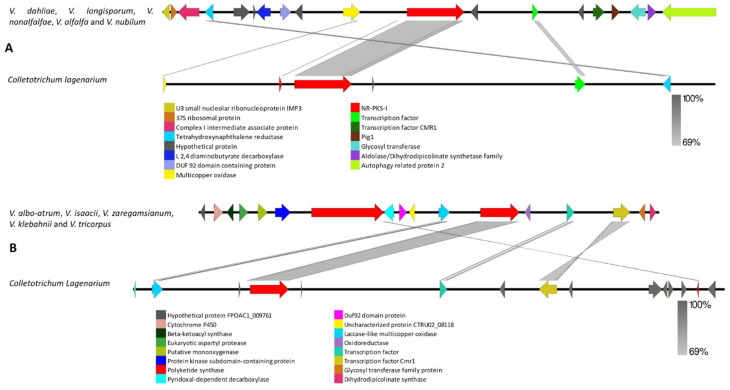
Comparison of putative cluster genes for *Verticillium* VNR-PKS-I-1 and the DHN melanin-producing PKS. (**A**) Putative VNR-PKS-I-1 cluster of *V. dahliae*, *V. longisporum*, *V. nonalfalfae*, *V. alfalfae* and *V. nubilum* compared to *Colletotrichum lagenarium* DHN melanin cluster (**B**) Putative biosynthetic cluster for *V. albo-atrum*, *V. isaacii*, *V. zaregamsianum*, *V. klebahnii* and *V. tricorpus* gene compared to *C. lagenarium* DHN melanin cluster. The arrowheads indicate the direction of transcription and the types of genes common to the two clusters being compared are displayed in the same color. Shaded lines represent similarities between nucleotide sequences.

**Figure 5 biology-11-01252-f005:**
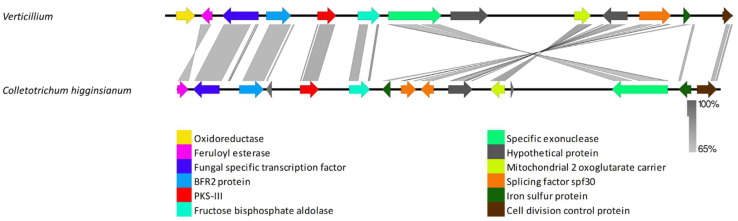
Comparison of *Verticillium* genes flanking PKS-III that are closely related homologs of PKS-III gene cluster from *Colletotrichum higginsianum*. The arrowheads indicate the direction of transcription and the types of genes common to the two clusters being compared are displayed in the same color. Shaded lines represent similarities between nucleotide sequences. This highly conserved cluster was found in all *Verticillium* genomes examined.

**Figure 6 biology-11-01252-f006:**
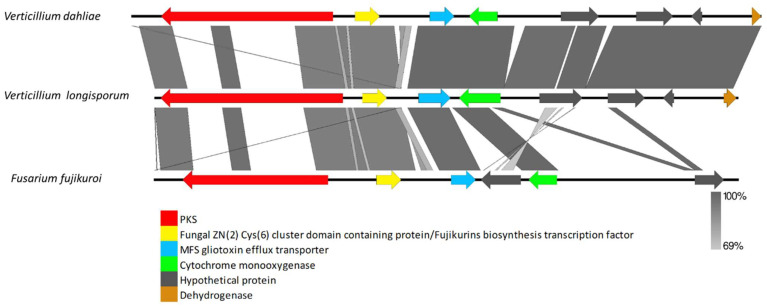
Comparison of *V. dahliae* and *V. longisporum* genes flanking VRPKS-I-8 PKSs that are closely related homologs of fujikurin synthase gene cluster from *Fusarium fujikuroi*. The arrowheads indicate the direction of transcription and the types of genes common to the two clusters being compared are displayed in the same color. Shaded lines represent similarities between nucleotide sequences.

**Figure 7 biology-11-01252-f007:**
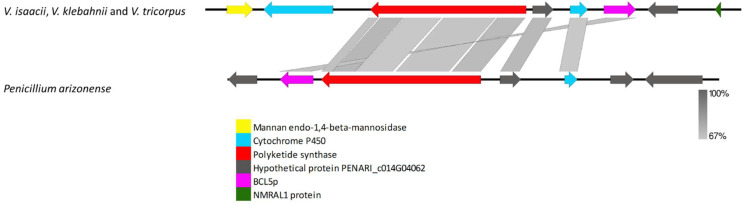
Comparison of putative cluster genes for VRPKS-I-20 locus found in the genomes of V. isaacii, V. klebahnii and V. tricorpus and an unknown PKS cluster from *Penicillium arizonense*. The arrowheads indicate the direction of transcription and the types of genes common between the two clusters being compared are displayed in the same color. Shaded lines represent similarities between nucleotide sequences.

**Figure 8 biology-11-01252-f008:**
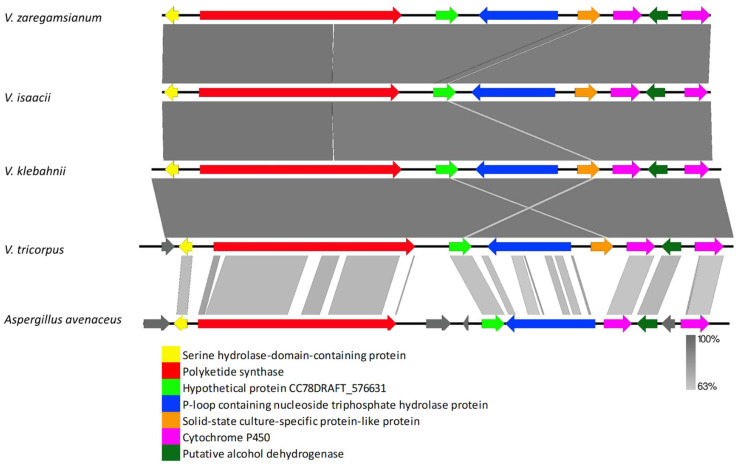
Comparison of putative cluster genes for VRPKS-I-6 cluster found in *V. zaregamsianum*, *V. isaacii*, *V. klebahnii* and *V. tricorpus* and burnettramic acid biosynthesis PKS cluster from *Aspergillus avenaceus*. The arrowheads indicate the direction of transcription and the types of genes common to the two clusters being compared are displayed in the same color. Shaded lines represent similarities between nucleotide sequences.

**Figure 9 biology-11-01252-f009:**
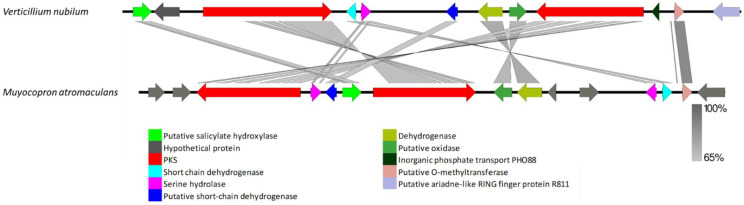
Comparison of putative cluster genes for *Verticillium nubilum* VNR-PKS-I-4 and the lijiquinone biosynthesis cluster of *Muyocopron atromaculans*. The arrowheads indicate the direction of transcription and the types of genes common to the two clusters being compared are displayed in the same color. Shaded lines represent similarities between nucleotide sequences.

**Figure 10 biology-11-01252-f010:**
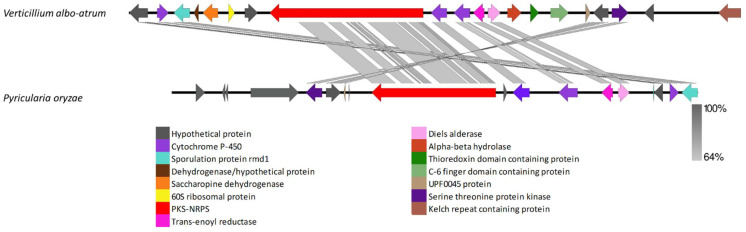
Comparison of putative cluster genes for *Verticillium albo-atrum* VRPKS-I-3 and *P. oryzae* cytochalasan clusters. The arrowheads indicate the direction of transcription and the types of genes common to the two clusters being compared are displayed in the same color. Shaded lines represent similarities between nucleotide sequences.

**Figure 11 biology-11-01252-f011:**
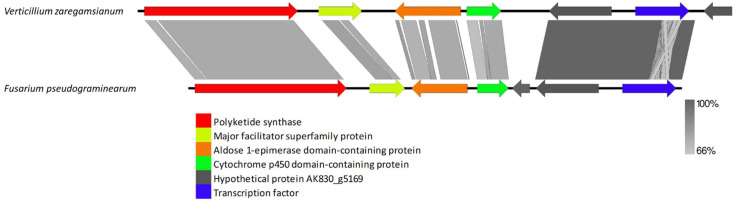
Comparison of putative cluster genes for *Verticillium zaregamsianum* VRPKS-I-7 cluster and fusarielin biosynthesis gene cluster from *Fusarium pseudograminearum*. The arrowheads indicate the direction of transcription and the types of genes common to the two clusters being compared are displayed in the same color. Shaded lines represent similarities between nucleotide sequences.

**Figure 12 biology-11-01252-f012:**
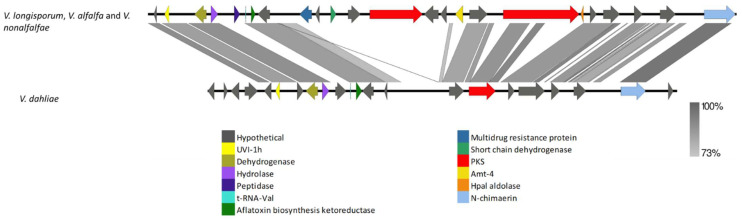
Comparison of VRPKS-I-11 gene cluster among pathogenic strains of *Verticillium*. VRPKS-I-11 was absent in *V. dahliae* but was present in *V. longisporum*, *V. alfalfae*, and *V. nonalfalfae*.

**Table 1 biology-11-01252-t001:** Isolates numbers and genome sequence information for the species used in this study.

Species	Isolate Number	GenBank Accession Number	References
*V. dahliae*	VdLs.17	ABJE00000000	[42]
*V. albo-atrum*	PD747	NMXJ00000000	[43]
*V. isaacii*	PD618	NMXN00000000	[43]
*V.alfalfae*	VaMs.102	ABPE00000000	[42]
*V. nonalfalfae*	VnAa140/PSU140/NRRL 66861	RBVV00000000	[44]
*V. nubilum*	PD621	NMXI00000000	[43]
*V. zaregamsianum*	PD739	NMXM00000000	[43]
*V. longisporum*	VL1	JAETXT000000000	[45]
*V. klebahnii*	PD401	NMXL00000000	[43]
*V. tricorpus*	MUCL 9792	JPET00000000	[46]

**Table 2 biology-11-01252-t002:** Comparison of the size (in amino acid residues) and domain structure of the *Verticillium* NR-PKS-I proteins with that of the top BLASTp hit in the NCBI database.

*Verticillium* NR-PKS-I Used as a Query	Size	Domain Organization *	Information Regarding the Top BLASTp Hit			
			Accession Number	Species	Size (aa)	Domain Organization *
**VNR-PKS-I-1**						
All *Verticillium* genomes	2164–2189	KS-AT-ACP-ACP-TE	XP_038751444	*Colletotrichum karsti*	2179	KS-AT-ACP-ACP-TE
**VNR-PKS-I-2**						
*V. zaregamsianum and V. klebahnii*	2124–2126	KS-AT-ACP-ACP-TE	RYP01440	*Monosporascus ibericus*	2108	KS-AT-DH-ACP-ACP-TE
**VNR-PKS-I-3**						
*V. alfalfa*, *V. nonalfalfae* and *V. longisporum*	1092–1911	KS-AT-ACP-ACP-TE	XP_028463040	*Sodiomyces alkalinus*	1919	KS-AT-ACP-ACP-TE
*V. longisporum*	1259	KS-AT-ACP-ACP-TE-**MFS**	KND89106	*Tolypocladium ophioglossoides*	2147	KS-AT-ACP-TE
*V. dahliae*	2158	KS-AT-ACP-ACP-TE-**MFS**	XP_028463040	*Sodiomyces alkalinus*	1919	KS-AT-ACP-ACP-TE
**VNR-PKS-I-4**						
*V. nubilum*	2578	KS-AT-ACP-MT-**NAD binding 4**	KAH7133939	*Dactylonectria macrodidyma*	2280	KS-AT-ACP-MT

* KS, AT, DH, ACP, NAD_binding_4, and TE correspond to ketosynthase, acyltransferase, dehydratase, acyl carrier protein, male sterility protein, and thioesterase domains, respectively.

**Table 3 biology-11-01252-t003:** Comparison of the size and domain structure of the *Verticillium* R-PKS-I proteins with top BLASTp hits in the NCBI database.

*Verticillium* R-PKS-I Used as a Query	Size	Domain Organization *	Information Regarding the Top BLASTp Hit			
			Accession	Name	Size (aa)	Domain Organization *
**VRPKS-I-1 (PKS-NRPS)**						
*V. dahliae, V. zaregamsianum, V. alfalfae* and *V. longisporum*	3750–4123	KS-AT-DH-ER-KR-C-A-ACP	KAF2874833	*Massariosphaeria phaeospora*	3929	KS-AT-DH-**MET**-ER-KR-C-A-ACP
**VRPKS-I-2 (PKS-NRPS)**						
*V. isaacii*, *V. klebahnii*, *V. tricorpus* and *V. albo-atrum*	3800–4429	KS-AT-DH-ER-KR-ACP-C-A-ACP	KAF4886526	*Colletotrichum fructicola*	3989	KS-AT-DH-ER-KR-ACP-C-A-ACP
**VRPKS-I-3 (PKS-NRPS)**						
*V. albo-atrum*	4677	KS-AT-DH-ER-KR-C-A-ACP-**NN** *	XP_036576988	*Colletotrichum truncatum*	4127	KS-AT-DH-ER-KR-C-A-ACP
**VRPKS-I-4 (PKS-NRPS)**						
*V. nubilum*	3987	KS-AT-DH-MET-ER-KR- C-A-ACP	KAI1485991	*Biscogniauxia mediterranea*	3976	KS-AT-DH-MET-ER-KR-C-A-ACP
**VRPKS-I-5 (PKS-NRPS)**						
*V. albo-atrum*, *V. zaregamsianum, V. isaacii* and *V. klebahnii*	3912–4017	KS-AT-DH-**MET**-ER-KR-C-A-ACP	TGO20046	*Botrytis tulipae*	4020	KS-AT-DH-ER-KR-C-A-ACP
*V. tricorpus*	4131	KS-AT-DH-MET-ER-KR-C-A-ACP-**NAD_binding_4**	KAI1485991	*Biscogniauxia mediterranea*	3976	KS-AT-DH-MET-ER-KR-C-A-ACP
**VRPKS-I-6 (PKS-NRPS)**						
*V. tricorpus*, *V. zaregamsianum, V. isaacii* and *V. klebahnii*	4028	KS-AT-DH-ER-KR-ACP-C-A-ACP	KAE8150730	*Aspergillus avenaceus*	4038	KS-AT-DH-**MET**-ER-KR-ACP-C-A-ACP
**VRPKS-I-7**						
*V. zaregamsianum*	2690	KS-AT-DH-MET-ER-KR	KAH8169122	*Sarocladium implicatum*	2675	KS-AT-DH-MET-ER-KR
**VRPKS-I-8**						
*V. dahliae* and *V. longisporum*	2506–2574	KS-AT-DH-ER-KR	ALQ32784	*Fusarium babinda*	2460	KS-AT-DH-ER-KR
**VRPKS-I-9 (PKS-NRPS)**						
*V. alfalfae*, *V. nonalfalfae*, *V. albo-atrum*, *V. tricorpus*, *V. isaacii* and *V. klebahnii*	3644–4064	KS-AT-DH-ER-KR-ACP-C-A-ACP	XP_044717051	*Hirsutella rhossilie*	3845	KS-AT-DH-ER-KR-ACP-C-A-ACP
**VRPKS-I-10**						
*V. nubilum*	1703	KS-AT-KR-**ACP**	KAH8906691	*Coniochaeta sp.*	2576	KS-AT-**DH-ER**-KR
**VRPKS-I-11(PKS-NRPS)**						
*V. alfalfae*	3529	KS-AT-DH-ER-KR-C-A-ACP	KAB2573439	*Lasiodiplodia theobromae*	3721	KS-AT-DH-ER-KR-**ACP**-C-A-ACP
*V. dahliae*	1113	KS-AT-DH	KAF4511962	*Ophiocordyceps sinensis*	1029	KS-AT-DH
*V. nonalfalfae*	3858	KS-AT-DH-ER-KR-ACP-C-A-ACP	KFH45364	*Acremonium chrysogenum*	3926	KS-AT-DH-ER-KR-ACP-C-A-ACP
*V. longisporum* contig	2672	KS-AT-DH-ER-KR-ACP-C	KFH45364	*Acremonium chrysogenum*	3926	KS-AT-DH-ER-KR-ACP-C-**A-ACP**
**VRPKS-I-12 (PKS-NRPS)**						
*V. isaacii*, *V. klebahnii* and *V. tricorpus*	3943–3982	KS-AT-DH-MET-ER-KR-ACP- C-A-ACP-**NAD_binding_4**	XP_028469272	*Sodiomyces alkalinus*	3890	KS-AT-DH-MET-ER-KR-ACP- C-A-ACP-
*V. zaregamsianum* and *V. albo-atrum*	3891–3972	KS-AT-DH-MET-ER-KR-ACP-C-A-ACP	XP_028469272	*Sodiomyces alkalinus*	3890	KS-AT-DH-MET-ER-KR-ACP-C-A-ACP
**VRPKS-I-13**						
*V. isaacii* and *V. klebahnii*	3080–3128	KS-AT-DH-ER-KR-ACP	KIL85244	*Fusarium avenaceum*	3197	KS-AT-DH-**MET**-ER-KR-ACP
*V. tricorpus* and *V. zaregamsianum*	3176–3209	KS-AT-DH-MET-ER-KR-ACP	KIL85244	*Fusarium avenaceum*	3197	KS-AT-DH-MET-ER-KR-ACP
**VRPKS-I-14**						
*V. zaregamsianum*, *V. tricorpus*, *V. isaacii* and *V. klebahnii*	2898–3161	KS-AT-DH-MET-ER-KR-ACP	XP_045957259	*Truncatella angustata*	3173	KS-AT-DH-MET-ER-KR-ACP
**VRPKS-I-15**						
*V. nubilum* contig	2690	KS-AT-DH-MET-ER-KR	KAH7133936	*Dactylonectria macrodidyma*	2606	KS-AT-DH-MET-ER-KR-**ACP**
**VRPKS-I-16**						
*V. isaacii*, *V. albo-atrum*, *V. zaregamsianum*, *V. tricorpus*, and *V. klebahnii*	2575–2718	KS-AT-DH-MET-ER-KR-ACP	KAH7369281	*Plectosphaerella cucumerina*	2570	KS-AT-DH-MET-ER-KR-ACP
**VRPKS-I-17**						
*V. zaregamsianum*, *V. albo-atrum*, *V. isaacii*, *V. alfalfae*, V. *nonalfalfae*, *V. tricorpus* and *V. klebahnii*	2158–2468	KS-AT-DH-ER-KR	OLN86478	*Colletotrichum chlorophyti*	2290	KS-AT-DH-ER-KR
*V.* nubilum, *V. dahliae* and *V. longisporum*	2268–2296	KS-AT-DH-ER-KR-**ACP**	OLN86478	*Colletotrichum chlorophyti*	2290	KS-AT-DH-ER-KR
**VRPKS-I-18**						
*V. isaacii*	2256	KS-AT-DH-ER-KR	POR36838	*Tolypocladium paradoxum*	2322	KS-AT-DH-ER-KR
*V. klebahnii* and *V. zaregamsianum*	2565–2857	**AMP**-KS-AT-DH-ER-KR	UJO18003	*Fulvia fulva*	2256	KS-AT-DH-ER-KR
**VRPKS-I-19**						
*V. longisporum*, *V. dahliae, V. zaregamsianum, Verticillium albo-atrum*, *V. klebahnii, V. isaacii*, *V. nonalfalfae, V. tricorpus*, and *V. longisporum*	2108–2291	KS-AT-DH-ER-KR-**ACP**	KAG6264852	*Claviceps purpurea*	2196	KS-AT-DH-ER-KR
*V. alfalfae* and *V. nubilum*	1524–2174	KS-AT-DH-ER-KR	KAG6041353	*Epichloe festucae*	2081	KS-AT-DH-ER-KR
**VRPKS-I-20**						
*V. isaacii* contig	3430	KS-AT-DH-ER-KR-ACP-**CYP120A1-CYPOR**	KAF4594413	*Ophiocordyceps camponoti-floridani*	2547	KS-AT-DH-**MET**-ER-KR
*V. klebahnii* contig	2354	KS-AT-DH-ER-KR-ACP	XP_022486633	*Penicillium arizonense*	2589	KS-AT-DH-**MET**-ER-KR-ACP
*V. tricorpus* contig	2475	KS-AT-DH-MET-ER-KR-ACP	XP_022486633	*Penicillium arizonense*	2589	KS-AT-DH-MET-ER-KR-ACP
**VRPKS-I-21**						
*V. zaregamsianum*, *V. klebahnii* and *V. nubilum*	2331–2532	KS-AT-DH-MET-ER-KR	KAF6826616	*Colletotrichum plurivorum*	2541	KS-AT-DH-MET-ER-KR-**ACP**
**VRPKS-I-22**						
*V. dahliae*, *V. alfalfae*, *V. nonalfalfae* and *V. longisporum*	2554–2635	KS-AT-DH-MET-ER-KR-ACP	ALQ32877	*Fusarium miscanthi*	2644	KS-AT-DH-MET-ER-KR-ACP
**VRPKS-I-23**						
*V. dahliae*, *V. longisporum*, *V. alfalfae*, *V. nonalfalfae*, *V. klebahnii*, *V. zaregamsianum* and *V. tricorpus*	2291–2402	KS-AT-DH-ER-KR-**ACP**	XP_028462492	*Sodiomyces alkalinus*	2401	KS-AT-DH-ER-KR
*V. nubilum*	2376	KS-AT-DH-ER-KR	XP_028462492	*Sodiomyces alkalinus*	2401	KS-AT-DH-ER-KR
*V. albo-atrum* contig	2651	KS-AT-DH-**MET**-ER-KR	XP_028462492	*Sodiomyces alkalinus*	2401	KS-AT-DH-ER-KR
*V. isaacii* contig	2347	KS-AT-DH-ER-KR	XP_028462492	*Sodiomyces alkalinus*	2401	KS-AT-DH-ER-KR
**VRPKS-I-24**						
*V. albo-atrum* contig	2453	KS-AT-DH-MET-ER-KR-**ACP**	KAF7552999	*Cylindrodendrum hubeiense*	2560	KS-AT-DH-MET-ER-KR
**VRPKS-I-25**						
*V. klebahnii*, *V. isaacii*, *V. zaregamsianum*, *V. albo-atrum* and *V. tricorpus*	2103–2358	KS-AT-DH-ER-KR-**ACP**	KFA46917	*Stachybotrys chartarum*	2393	KS-AT-DH-ER-KR
**VRPKS-I-26**						
*V. dahliae*, *V. longisporum*, *V. alfalfae*, *V. nonalfalfae*,	1748–2345	KS-AT-DH-KR-ACP	RYP50633	*Monosporascus* sp.	2358	KS-AT-DH-**MET**-**ER**-KR-ACP
*Verticillium albo-atrum*	2326	KS-AT-DH-KR-**TTL**	RYP50633	*Monosporascus* sp. mg162	2425	KS-AT-DH-**MET**-**ER**-KR-**ACP**
*V. zaregamsianum*	2349	KS-AT-DH-ER-KR-ACP	P0CU84	*Apiospora sphaerosperma*	2466	KS-AT-DH-ER-KR-ACP
*V. isaacii*, *V. nubilum*, *V. klebahnii* and *V. tricorpus*	2345–2350	KS-AT-DH-ER-KR	P0CU84	*Apiospora sphaerosperma*	2466	KS-AT-DH-ER-KR-**ACP**

* KS, AT, ACP, DH, KR, ER, MET, NAD_binding_4, AMP, CYP120A1, CYPOR and NN correspond to ketosynthase, acyltransferase, acyl carrier protein, dehydratase, ketoreductase, enoylreductase, methyltransferase, male sterility protein, AMP-binding enzyme/acyl-CoA synthetase, cytochrome P450 family and Non known domains, respectively. Boldfaced domains indicate when domain organizations between each ORF and its BLAST hit differed.

## Data Availability

The study did not report any data.

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
