# Peer review of "Genome-Based Analysis of Verticillium Polyketide Synthase Gene Clusters"

_biology, 2022, doi:10.3390/biology11091252_

Round 1
Reviewer 1 Report
I have read with interest the submitted manuscript “Genome-based analysis of Verticillium polyketide synthase gene clusters”. This manuscript identified 32 putative PKS genes in the genome of 10 Verticillium genus. The authors presented novel data and insights and have used adequate and appropriate techniques.
I have some minor comments that the authors could consider.
It is better to characterize the function of one or more Verticillium PKS genes in secondary metabolites synthesis by gene knockout or by citing literatures.
The species names and gene symbols should be italic.
In some places, the authors missed a "e" in dahliae.
Author Response
Response to Reviewer #1’ Comments
Dear reviewer,
Thank you for your time and effort spent reading our manuscript. Please note that all changes requested by you are highlighted throughout the manuscript.
- “…it is better to characterize the function of one or more Verticillium PKS genes in secondary metabolites synthesis by gene knockout or by citing literatures”.
Thank you for the good point. Making knock-out strains to check the function of the identified genes is something that we are planning to do in the near future. Please note that we have added a few sentences to the manuscript and cited previous work on the role of PKS genes in Verticillium as shown in highlighted text in section 4.2.
- “….the species names and gene symbols should be italic”.
We have gone through the manuscript very carefully and italicized all species names.
- The reviewer pointed out that in some places, the authors missed "e" in dahliae.
Sorry for the mistake! We checked the manuscript very carefully and corrected V. dahliae name in entire manuscript.
Reviewer 2 Report
I consider it to be a valuable study.
I make the following suggestions to the authors
· Please report the address Aria Dolatabadian
· “Verticillum, V. dahliae, Sclerotinia sclerotiorum, Brassica napus, V. longisporum, F. fujikuroi, V. albo-atrum, V. isaacii, V. tricorpus, V. zaregamsianum, V. klebahnii, V. alfalfa, V. nonalfalfae, V. isaacii, V. tricorpus, V. klebahnii, V. nubilum” should be written in italics in the text.
· Introduction page 2: “revise…secondary metabolites or natural products (SMs, NPs)”…I suggest “respectively”
· Introduction: Page 2. Please reise the sentence: "Ascomycetes contain more secondary metabolism genes than basidiomycetes, archeo-ascomycetes, chytridiomycetes, hemi-ascomycetes and zygomycetes [24]."
· Please revise the format of the References.
· Please revise in Discussion the 4.5. Some evidence of horizontal gene clusters
· I suggest reviewing the scope 5. Conclusions
Author Response
Response to Reviewer 2’ Comments
Dear reviewer,
Thank you a lot for your time and effort reading and commenting on our manuscript. Please note that all changes requested by you are now shown in track change option of word document throughout the manuscript.
- The reviewer asked that all the species names should be written in italics in the text.
Sorry for the mistake. We have gone through the manuscript and made sure that all the species names are in Italics.
- The reviewer suggested that in Introduction page 2, we should add ‘’respectively’’ to the end of following sentence: …secondary metabolites or natural products (SMs, NPs)”…”.
We have added ‘’respectively’’ as suggested as shown in highlight.
- The reviewer asked us to revise the following sentence in Page 2: "Ascomycetes contain more secondary metabolism genes than basidiomycetes, archeo-ascomycetes, chytridiomycetes, hemi-ascomycetes and zygomycetes’’.
We have revised the sentence as suggested by reviewer as shown in highlight in the text.
- The reviewer suggested to revise the format of the References.
Dear reviewer, the current format of the references is the acceptable format based on the journal’s guideline.
- The reviewer asked to revise in Discussion the 4.5.
We have revised this section and removed most of the parts related to horizontal gene transfer as shown in highlight in the section 4.5.
- The reviewer suggested to revise the conclusion part.
We have revised this section.
Reviewer 3 Report
The study in silico prediction and characterisation of putative secondary metabolite clusters in Verticillium species using a comparative genomics approach could shed light on the roles of these clusters and their evolution. There are a few minor revisions necessary to address. See attached file for specific comments and suggested revisions. My suggestions have highlighted in the document and linked to a number
1) Remember the scientific names in Latin always are italicized
2) I estimate that these lines do not bring information relevant to the introduction neither I could find a relationship with the object of study
3) Add the full name “Hybrid Polyketide Synthase-Nonribosomal Peptide Synthetase”
4) Switch 10-5 by 10-5
5) Suggest the ID gene of each Verticillium species should be added and keep the gene name as it was mentioned for the first time to avoid reader confusion, I recommend to use name by Shi-Kunne et al. as a reference. Remember to modify the names in the text
6) Unify the format of this paragraph with the document
7) I consider that this section does not have a solid support in the analysis carried out on this work and it is very speculative. Perhaps, additional data could validate or support these results.
8) I would eliminate these sentences, because I do not find its importance within this paragraph
9) Highlight the new identified clusters or new genes associated with identified clusters previously, could be interesting for the reader
10) I consider that the horizontal gene transfers in this section does not have a solid support in the analysis carried out in this work and it is very speculative
11) Please, reference this work, and check other statements or sentences because there are not some cited.

Author Response
Response to Reviewer #3’ Comments
Dear reviewer,
Thank you for your time and effort spent on reading and commenting on our manuscript. Please note that all changes requested by you are now highlighted throughout the manuscript.
- The reviewer asked that the species names must be italic.
Sorry for this mistake! We have gone through the manuscript very carefully and made sure that all species names are italicized.
- The reviewer pointed out that the following lines do not bring information relevant to the introduction: ‘’In addition, the expression of 15 dahlia's genes, putatively involved in pathogenicity, showed differential upregulation in the highly versus weakly aggressive isolates in response to plant extracts or after inoculation of potato leaf petioles [14]’’.
We’ve removed the above-mentioned lines from the introduction.
- The reviewer suggested to add the full name “HybridPolyketide Synthase-Nonribosomal Peptide Synthetase in section 2.2.
We have added the full name of PKS-NRPs to the section 2.2 as shown in highlight in the text.
- In section 2.5, the reviewer suggested to switch 10-5 by 10-5.
We have corrected this as shown in section 2.54 in highlight.
- The reviewer suggested to use gene name given by Shi-Kunne et al. as a reference.
We named all genes in the same order as they appeared in the genome through our genome analysis. We started from VRPKS-I-1 to VRPKS-I-28 for Verticillium reducing-PKS-I genes and VNR-PKS-I-1 to VNR-PKS-I-4 for Verticillium non-reducing PKS-I genes.
- The reviewer asked to unify the format of the section 3.2 with the document.
We have unified the format of this section as shown in highlight.
- The reviewer mentioned that he/she thinks that the secion 3.2 (Gene gain/loss) does not have a solid support in the analysis carried out on this work and it is very speculative.
We have revised this section so that we don’t claim on gene gain/loss event, but only kept some important information about missing of some PKS genes/clusters in closely related Verticillium species which we think is quite important. However, we know that more evidence is needed to prove the gene gain/loss event, which we are going to carry out in near future.
- The reviewer suggested to eliminate the following sentence from section 3.2: ‘’VRPKS-I-11 contained two PKS-NRPS genes’’.
We have removed the above-mentioned sentence from section 3.2.
- The reviewer suggested that highlighting the new identified clusters or new genes associated with identified clusters previously, could be interesting for the reader.
We have added a few sentences on the previously published work on some of the identified clusters in this study as shown in highlight in the sections 4.2 and 4.8.
- The reviewer suggested that the horizontal gene transfers in the section 4.5 does not have a solid support in the analysis carried out in this work.
We have revised this section and removed most of the parts related to horizontal gene transfer as shown in highlight in the section 4.5.
- The reviewer asked to add the reference for the following sentence in section 4.6: ‘’it has also been reported that mutation of either the transcription factor VdCmr1 or the VdPKS1 did not reduce virulence and both genes were not required for pathogenesis on tobacco and lettuce’’.
Sorry for missing this reference. We have added the reference for this statement as shown in highlight in the text.
Round 2
Reviewer 3 Report
I agree with modification